# Individual and Contextual Factors Associated with Classroom Teachers’ Intentions to Implement Classroom Physical Activity

**DOI:** 10.3390/ijerph20043646

**Published:** 2023-02-18

**Authors:** Gabriella M. McLoughlin, Hannah G. Calvert, Lindsey Turner

**Affiliations:** 1College of Public Health, Temple University, Philadelphia, PA 19140, USA; 2Implementation Science Center for Cancer Control, Brown School, Washington University in St. Louis, St. Louis, MO 63130, USA; 3College of Education, Boise State University, Boise, ID 83725, USA

**Keywords:** physical activity, schools, classrooms, implementation science, intention, implementation determinants

## Abstract

Classroom-based physical activity (CPA) is an evidence-based practice that improves student physical activity outcomes, but national data suggest implementation is insufficient in US classrooms. The purpose of this study was to examine individual and contextual factors associated with elementary school teachers’ intentions to implement CPA. We collected input survey data from 181 classroom teachers (10 schools; 98.4% participation among eligible teachers) across three separate cohorts to examine associations between individual and contextual constructs and future CPA implementation intentions. Data were analyzed using multilevel logistic regression. Individual-level characteristics of perceived autonomy for using CPA, perceived relative advantage/compatibility of CPA, and general openness to educational innovations were positively associated with intentions to implement CPA (*p* < 0.05). Teacher perceptions of contextual factors such as administrator support for CPA were also associated with implementation intentions. This study adds to prior evidence about the importance of theoretically determined constructs for understanding behavioral intentions among front-line implementers such as classroom teachers. Additional research is needed to evaluate interventions designed to change malleable factors, including teachers’ perceptions, as well as changing school environments so that teachers perceive more autonomy to use CPA and have the training and resources that build skills for implementation.

## 1. Introduction

Physical activity (PA) has important health benefits, and it is recommended that children accrue at least 30 min of activity during school time [1,2]. Classroom PA (CPA) includes two approaches, both of which are implemented by classroom teachers: activity breaks, which are brief breaks from instruction for movement, and active lessons, which involve the integration of movement with academic content [3,4]. Two systematic reviews document the benefits from CPA [5,6], making CPA an evidence-based practice for improving PA and educational outcomes [3,4,5,6,7,8].

Despite the benefits of CPA, nationally representative studies suggest that fewer than half of US elementary school teachers implement CPA regularly [9,10,11]. Teachers perceive many benefits of CPA, such as student enjoyment, improved attention, and improved classroom climate [12,13,14,15,16]. However, teachers also report many barriers to implementing CPA such as lack of time, resources, and leadership support [12,13,14,15]. Organizational leadership is an important determinant of the implementation of many evidence-based practices [17,18], including CPA [19,20,21]. The availability of resources, perceived benefits of CPA, and perceived implementation climate have all been associated with teachers’ CPA implementation [22,23,24]. Determinants of behavior change include individual-level characteristics such as knowledge, attitudes, and self-efficacy [25,26,27]. These are also evident for CPA behaviors, as teachers’ implementation is positively associated with self-efficacy [28,29] and perceptions about norms and administrator support for CPA [23].

### 1.1. Theoretical Grounding for Exploring Teachers’ Intentions to Implement

#### 1.1.1. Contextual Implementation Theories and Frameworks

A teacher’s implementation of CPA is an individual-level behavior—albeit one that impacts all students in the classroom—and thus a social-ecological approach [30] that acknowledges the interplay between individuals and their environment is useful in conceptualizing teachers’ behavior as front-line implementers. Given the literature documenting the influence of organizational culture and leadership on teachers’ behaviors, determinants of implementation [31] are best conceptualized with a systems perspective that considers various levels (e.g., individual, organizational), and interactions across levels. Common determinants include organizational support, social relations, and leadership [32,33,34]. Although teachers within a school share a similar context, their perceptions of those environments may differ.

One of many frameworks for conceptualizing the various implementation determinants is the Consolidated Framework for Implementation Research (CFIR; [26,35]), which comprises several domains of determinants for implementing evidence-based practices. Such domains include innovation characteristics (i.e., aspects of CPA as an intervention), outer setting (i.e., factors outside the organization), inner setting (i.e., factors within the organization), characteristics of individuals (i.e., skills/knowledge, self-efficacy, and motivation), and implementation process (i.e., planning, engaging, for implementation). Through the last several years of school-based research, the domains that are most prevalent as determinants of the implementation of CPA are the innovation characteristics, specifically the complexity and evidence base of an evidence-based practice; the “inner setting” that captures an array of organization (i.e., school)—level determinants; and characteristics of individuals, specifically motivation and openness to new innovations [34,36]. The CFIR has shown high utility in a variety of implementation studies in healthcare and education [34,36], warranting the use of several constructs in the present study.

#### 1.1.2. Individual-Level Behavior Theories and Frameworks

For many years, the Theory of Reasoned Action (TRA; [37]) and its extension as the Theory of Planned Behavior [38] have been used to explain individual-level health behaviors, acknowledging that a crucial proximal outcome before behavior change is a behavioral intention, or the plan to engage in a behavior. Intentions are determined by attitudes toward a behavior, perceived social norms regarding that behavior, and the perceived control of one’s own behavior. The proximal outcome of behavioral intention—while not a perfect predictor of subsequent behaviors—is among the strongest predictors of behavior [39,40], including in applications with teachers [41,42]. As applied to teachers’ intentions to implement CPA, attitudes could include outcome expectancies such as perceived impacts on student behavior, attention, and academic outcomes. Perceived social norms include perceptions about whether administrators support CPA, and whether other teachers in the school are engaging in CPA. Lastly, perceived behavioral control is a prominent element of many behavior theories and predicts teachers’ CPA implementation behaviors [22,43].

Motivation is a crucial determinant of implementation outcomes, and our current work, as with several prior CPA studies, applies Self-Determination Theory (SDT; [44,45,46,47]). As a motivation theory, SDT emphasizes three key components: the intrinsic motivation of autonomy (i.e., control over one’s behaviors), competence (i.e., perceived ability to carry out a behavior), and relatedness (i.e., feeling a sense of connection with others). Specifically, prior research has demonstrated that teachers who have greater perceived autonomy and competence are more likely to implement CPA [47]. Prior research also highlights that school leaders (i.e., administrators) are a key predictor of teacher behavior, and they often are “gatekeepers” of a teacher’s autonomy [43]. As such, promoting autonomy-supportive leadership practices in school settings is important, and we view this as an integral component to consider as a potential determinant of CPA. The use of SDT and TRA in the current study provided a robust way to assess constructs of “high- and mid-level leaders” and “innovation deliverers” that sit within the Characteristics of Individuals domain in CFIR. As such, using a combination of these three frameworks facilitates a multi-level understanding of key predictors of CPA.

### 1.2. Study Purpose

The purpose of this study was to examine individual and contextual factors associated with elementary school teachers’ intentions to implement CPA. We hypothesized that perceived benefits (compatibility and relative advantage), in addition to perceived competence, administrator support, norms, and current practices would predict intentions, and that perceived barriers would be negatively associated with intentions.

## 2. Materials and Methods

This study utilizes survey data gathered at baseline of a project to study teachers’ implementation of CPA. The outcome explored here was intention—that is, plans to implement CPA during the subsequent school year. The study was conceptualized using a combination of SDT, TRA, and CFIR to ensure a balance of individual and contextual determinants of CPA adoption. Ten public elementary schools in southwestern Idaho were recruited, based on proximity to the university (all within 70 miles) and interest in the project. School-level demographic data were collected from administrative sources, including school-level poverty (percentage of students eligible for free/reduced-priced meals), locale, and number of students.

### 2.1. Procedure

The project was conducted over three years, with 3 schools participating in 2015–2016, 2 schools in 2016–2017, and 5 schools in 2017–2018. In each of the three years, this project began with the delivery of a professional development workshop in the final weeks of summer before the school year began. Apart from 3 teachers absent due to illness, all classroom teachers (98.4%) at the participating schools attended the workshop and provided informed consent to engage in the study. In each of the three cohorts, the summer workshop lasted 4 h and presented information about CPA, demonstrations of how to conduct CPA, and guided support for teachers to practice leading CPA. Teachers received the Energizers curriculum [48] and access to Go Noodle Plus^®^ during/after the training. Teachers were encouraged to provide CPA each school day and were asked to complete tracking logs during the fall semester. The tracking logs varied from year to year, so the implementation data cannot be combined; analyses reported here only pertain to intentions, not behaviors. A total of 181 teachers participated, and each received a $300 stipend.

Demographic characteristics of schools and teachers are provided in Table 1. Among the 181 teachers, grades taught ranged from Kindergarten through 6th grade, with four in other circumstances (resource, Title I). Most teachers were female (90.6%), with between 1 and 39 years of teaching experience, with nearly half (40.3%; n = 73) in the first 5 years of their career.

### 2.2. Instrumentation

In each of the three cohorts, a one-page survey was administered at the beginning of the workshop and a six-page survey was administered at the end of the workshop. The first survey assessed each teacher’s prior use of CPA with two questions: “*How often do you use movement breaks such as Energizers in your classroom*?” and “*How often do you use active lessons (keeping students moving during instruction)?*” with options of never, rarely, sometimes, often, and very often. A third item examined the extent to which teachers perceived their school had “*encouraged classroom teachers to promote physical activity in the classroom*” with responses of never, not much, somewhat, quite a lot, and very much.

The survey at the end of the workshop addressed teachers’ CPA intentions and constructs hypothesized to be related to intentions, including perceived autonomy and competence, benefits, barriers, administrator support, and perceived norms. Table 2 presents the exact wording of each survey item, with means and standard deviations for items and scales, factor loadings for each item, and coefficient alpha for each scale. Each construct, and its associated theoretical framework, is briefly described below.

### 2.3. Contextual Implementation Determinants (CFIR)

#### 2.3.1. Relative Advantage and Compatibility (Innovation Characteristics)

Previously used items assessing teachers’ perceptions of the relative advantage of CPA and compatibility with instructional practices were included [24]. Responses were made on a six-point scale from 1 = strongly disagree to 6 = strongly agree.

#### 2.3.2. Perceived Barriers (Innovation Characteristics)

The Exercise Beliefs and Barriers Scale [49] was used to guide the identification of complexity, and themes noted in CPA research [12,13,14,16,23,50] guided the adaptation and development of items to address barriers (e.g., time, student reactions, academic implications). One item previously used elsewhere [24] assessed perceived complexity of CPA. Responses were made on a six-point scale from 1 = strongly disagree to 6 = strongly agree.

#### 2.3.3. Organizational Climate (Inner Setting)

Organizational climate was measured with subscales from the Organizational Climate Description Questionnaire [51], including six items to assess collegial behavior, and four items to assess restrictive behavior. Responses were made on a five-point scale where 1 = never occurs, 2 = rarely occurs, 3 = sometimes occurs, 4 = often occurs, and 5 = very frequently occurs.

#### 2.3.4. Openness to Educational Innovation (Characteristics of Individuals)

This scale included four items used in prior work assessing teachers’ implementation of CPA [24]. Responses were made on a six-point scale from 1 = strongly disagree to 6 = strongly agree.

### 2.4. Individual-Level Constructs (TRA and SDT)

#### 2.4.1. Perceived Norms for CPA (TRA)

This was measured with a single item asking teachers’ perceptions of the frequency of CPA use among other teachers in the school. This item was developed by the researchers to assess normative beliefs about CPA at the school. Responses were made on a six-point scale from 1 = strongly disagree to 6 = strongly agree.

#### 2.4.2. Principal Support for CPA (SDT)

This was measured with a single item from the Perceived Autonomy Support–Work Climate Questionnaire [52] adapted to be specific to CPA. Responses were made on a six-point scale from 1 = strongly disagree to 6 = strongly agree.

#### 2.4.3. Perceived Competence for CPA (SDT)

Three items were selected from the Perceived Competence Scale [53]. The wording was modified to be specific to CPA (i.e., “*I feel confident in my ability to manage my diabetes*” was changed to “*I feel confident in my ability to occasionally lead activity breaks in my classroom*.”). In years 1 and 2 these items used a five-point response metric (1 = not at all true to 5 = very true), which changed to a six-point metric (1 = strongly disagree to 6 = strongly agree) in year 3. To harmonize across cohorts, the five-point metric was converted to a six-point metric (1 = 1; 2 = 2.5; 3 = 3.5; 4 = 4.75; 5 = 6). Preliminary analyses (Table 3) compared means across cohorts to confirm comparability over time.

#### 2.4.4. Perceived Autonomy for CPA (SDT)

Three items were selected from the Basic Psychological Needs Satisfaction and Frustration Scale [54] with modification for CPA. For example, “*I feel like I am in charge of my exercise program*” was changed to “*I am in charge of how to incorporate PA into my classroom*”. In years 1 and 2, these items used a five-point response metric that changed to a six-point metric in year 3; responses on the five-point metric were converted (1 = 1; 2 = 2.5; 3 = 3.5; 4 = 4.75; 5 = 6) to calculate the scale score.

#### 2.4.5. Perceived Autonomy Support (SDT)

Six items from the Perceived Autonomy Support Work Climate Questionnaire [52] were used to assess teachers’ perceptions of the extent to which their principal generally supports autonomy for a variety of educational practices. Responses were made on a five-point scale from 1 = not at all true to 5 = very true.

#### 2.4.6. Intention to Provide CPA

Each teacher’s behavioral intention to provide CPA was assessed with a single item asking for extent of agreement with the statement “*I definitely plan to try using activity breaks in my classroom*”. Responses were made on a six-point scale from 1 = strongly disagree to 6 = strongly agree.

### 2.5. Data Analysis

After examining the characteristics of the schools and teachers, descriptive statistics were computed, including means and standard deviations for each item and construct. Coefficient alpha was computed for each scale (all were >0.7). In addition, the item-to-total correlation (Table 2) provides an indication of whether each item contributes to the overall scale score, with all items being acceptable at more than 0.40. Scale scores were created by taking the average of responses to all included items. Average responses on each construct were compared across schools (Table 3) and tested for differences with ANOVA in Stata while accounting for clustering of teachers within school. Thereafter, correlations among constructs were calculated (Table 4).

### 2.6. Exploration of Perceived Norms and Actual Norms

An exploratory analysis investigated the accuracy of teachers’ perceptions of CPA norms at their school. A single-variable estimate of percent CPA use (active lessons or movement breaks) prior to the professional development workshop was created using the two questions, “*How often do you use movement breaks such as Energizers in your classroom*?” and “*How often do you use active lessons (keeping students moving during instruction)?*” For each teacher, if they responded “often” or “very often” to either question, they were coded as 1 = a CPA implementer, versus 0 = not a regular CPA implementer. The number of teachers scored as 1 was summed at each school and was divided by the number of teachers at that school to create a variable reflecting percentage of teachers implementing CPA within each school (considered to be a proxy for “actual norms”). We explored how teachers’ perceptions of CPA norms related to the proxy of actual norms in each school using a linear regression model that accounted for clustering of teachers within schools.

### 2.7. Multilevel Regression Analyses

Because of the relatively small number of clusters (n = 10 schools), a one-level regression model was used to explore the relationship between behavioral intention and explanatory constructs (Table 5). An approach recommended by Cameron et al. [55] was used, whereby models were estimated using maximum likelihood with robust standard errors and the sandwich estimator to adjust both standard errors and model fit for clustering effects. This is recommended as a rigorous but practical solution to handling bias when analyzing data with a small but meaningful number of clusters. Because of the downward bias the sandwich estimator may have on standard errors when analyzing data with a small (<30) number of clusters [56], *p*-values for model estimates were further adjusted using a t-distribution and degrees of freedom equal to the number of clusters minus 1, as well as a lower alpha of 0.025. The regression model was calculated in MPlus.

Teachers’ responses on the item to assess intentions were skewed towards the higher end of the response metric and were thus deemed unsuitable for linear regression. Instead, the outcome was dichotomized, and logistic regression was used, comparing teachers with a response of “strongly agree” (n = 116, 64.1%) versus those who disagreed or agreed only somewhat or moderately that they planned to implement CPA (n = 63, 34.8%; 2 teachers had missing data on this item). Models included all of the survey predictor variables, as well as teacher gender, grade taught, and number of years of teaching, Each teacher’s prior use of activity breaks and active lessons were entered as binary predictors in the regression model. The question “*How often do you use movement breaks such as Energizers in your classroom*?” was collapsed into a variable where responses of often and very often (combined, 45.8% of teachers) were coded as 1, and responses of never (1.7%), rarely (10.7%), or sometimes (41.8%) were coded as 0. The question “*How often do you use active lessons (keeping students moving during instruction)?*” was similarly collapsed as often or very often (combined, 21.3% of teachers) versus never (0.6%), rarely (22.5%), or sometimes (55.6%). School-level poverty was also included as a control variable, based on the percentage of students eligible for free/reduced-price meals, and was coded as 0 = <33%, 1 = 33% to <66%, and 2 = >66%.

## 3. Results

Mean scores on most constructs varied across schools (Table 2); those pertaining to school context (i.e., perceived norms, principal support, organizational climate) varied more across schools than teacher characteristics such as perceived competence, which did not vary much and was relatively high on average; scores for perceived barriers were relatively low (Table 3).

### 3.1. Relationship between Perceived Norms and Actual Norms

Exploratory analysis revealed a significant positive relationship between teacher-perceived norms and our calculation of the percentage of teachers who reported using CPA at each school. In a regression conducted at the teacher level and accounting for school clustering, school-level “actual norms” was positively associated with each teacher’s perception of school norms (β = 0.462, *p* < 0.001; results not shown in tables).

### 3.2. Correlations

Table 4 presents bivariate correlations among constructs. Examination of the correlation matrix indicated many statistically significant associations among constructs, but none were above 0.60, which could pose collinearity issues when simultaneously used as predictors in regression models.

### 3.3. Multilevel Regression Model: Predictors of Teacher Intentions to Implement CPA

In the multilevel logistic regression model (Table 5), several constructs significantly predicted teachers’ intentions to implement CPA. Teacher gender, years of teaching experience, and grade taught did not relate significantly to intentions. Teachers’ prior use of CPA was not associated with intentions; however, several individual and contextual factors were associated with the outcome. Perceived autonomy (OR = 2.99, *p* = 0.036) was positively related to teachers’ CPA intentions, as was the perceived advantage/compatibility of CPA (OR = 8.62, *p* = 0.006). Intentions were significantly higher among teachers who were more open to educational innovations (OR = 2.52, *p* = 0.044), and among teachers who perceived that their principal supported CPA (OR = 3.03, *p* = 0.036). The model fit well, as reflected by McFadden’s pseudo R^2^ of 0.507.

## 4. Discussion

We explored individual and contextual factors related to classroom teachers’ intention to implement CPA. Similar to other research exploring intentions to implement PA programming among intermediaries in community service organizations [57], we find that an array of individual and contextual characteristics—including perceptions about CPA, as well as aspects of the organizational context—were associated with intentions. Individual-level characteristics of perceived autonomy were positively related to intentions, aligning with several theories [25,38,45] and prior CPA research showing that intrinsic motivation is associated with teachers’ intentions to implement educational innovations [58,59]. Vazou and colleagues [59] showed that autonomous motivation positively predicted teachers’ engagement with fitness programming. Recent work has highlighted the importance of self-efficacy—similar to competence—in explaining teachers’ CPA implementation [29]. The current work did not find that perceived competence significantly explained implementation, despite the robust body of evidence showing the importance of teacher self-efficacy when implementing innovations in their classrooms [22,24,28]. However, our results showing the importance of autonomy, a motivational component articulated in Self Determination Theory, are relatively novel.

Our work reflects that of Webster and colleagues [24], who found greater self-reported CPA implementation among teachers who perceived benefits of CPA, and for whom CPA was compatible with their teaching philosophies. We also found that perceived benefits of CPA in terms of relative advantage/compatibility were strongly associated with intentions. Although these perceptions were generally quite high among teachers in our study, this construct was still predictive of intentions, despite the relatively limited variability. This highlights the importance of informing teachers about the relative advantage of CPA for improving student outcomes. Teachers’ perceptions that CPA is beneficial and compatible with their instructional practices warrant future professional development efforts, such as providing logistical strategies for integrating CPA into their routines (e.g., including CPA in lesson planning). The constructs of relative advantage and compatibility are both key elements of the Diffusion of Innovations theory [60], and the current results add to prior work showing the importance of these characteristics as determinants of the implementation of innovations in schools. As noted by Webster and colleagues [24] in their paper presenting the measure that we used to assess advantage/compatibility, their survey items assessing relative advantage did not compare to a concrete alternative such as asking about perceptions that CPA will “increase my students’ academic performance more than teaching sedentary lessons with students at their desks”. We used this exact wording for one item on our advantage scale and found that it held together well with other items and was significantly associated with intentions. As such, we agree that the explicit comparison to a concrete alternative is worthwhile when examining relative advantage.

In addition to the importance of CPA advantage and compatibility, another construct that we found to be significantly associated with teachers’ intentions was a broader openness to educational innovations, a construct also found by Webster and colleagues to predict CPA implementation [24]. This construct was positively associated with advantage/compatibility (r = 0.33, *p* < 0.001), but not overlapping to the point where it did not make an additional contribution to explaining intentions. This provides additional insights for CPA promotion, by encouraging classroom teachers to generally be more open to implementing new practices. The construct of attitudes toward evidence-based practices has been explored more widely in health services settings [61] but such attitudes also impact implementation behaviors among educators [62]. The exploration of such constructs could facilitate a greater understanding of how to increase teachers’ openness to implementing evidence-based practices.

Prior work has found that perceived barriers are associated with less use of CPA [18,23]; we found a negative but not statistically significant association with intentions (OR = 0.35, *p* = 0.093). Perceived barriers were negatively associated with relative advantage/compatibility (*r* = −0.45) and were negatively associated with constructs such as perceived competence, autonomy, and principal support. Those variables may be stronger in explanatory power, and teachers may perceive CPA to be less feasible (i.e., more impacted by barriers) if they lack the environmental support, resources, and training that allow them to feel competent to use CPA. Developers of CPA curricula and implementation supports should consider environmental facilitators and proactively making administrator support evident.

Lastly, we note the importance of perceived administrator support for CPA, similar to prior CPA research [15,16,19,24,50]. There is a wide recognition of the importance of system-level leadership for innovations to be successfully implemented in service organizations, and a growing interest in the potential for interventions to mobilize leadership that supports the implementation of innovations [63,64]. Such work in K−12 school settings will be valuable for understanding how to change organizational determinants of teachers’ attitudes and behaviors such as principal support for CPA and the encouragement of teachers’ autonomy to implement CPA in ways that work best for them. Additionally, the importance of tailoring CPA implementation strategies to the individual based on their role within the school (e.g., classroom teacher, principal, or physical education teachers) is appropriate, as individuals in these roles can perceive different barriers to implementation [65].

A novel element of the current work is the design that involves sampling nearly all teachers in a relatively small number of schools, rather than recruiting across a larger number of schools but with lower response rates from teachers within each school. For example, prior CPA work utilized data from 201 teachers at 79 elementary schools in South Carolina [24], 205 teachers (19.5% of potential respondents) at 20 schools in Texas [29], and 365 teachers (84.7% of potential responders) at 24 schools in California [23]. The similar results across the current and prior studies provide encouraging evidence of the combined importance of individual and contextual factors in understanding CPA implementation. Because the current study included almost all teachers from 10 schools, it was possible to explore the association between teachers’ perceptions about CPA norms at their school and the self-reported practices of other teachers, which we considered as an indication of “actual norms”. The association was modest, and perceived norms were not significant in the multivariate model, whereas other perceptions were associated with intentions.

### Limitations

Although this study is novel in collecting data on nearly all teachers at a sample of schools, the schools were selected based on administrators’ willingness to participate in a research project on CPA. Consequently, these results may not be generalizable to all schools. We also provided direct remuneration to teachers given that they were participating in professional development designed to improve learning outcomes in their students, which may have changed their perceptions about CPA. Second, although our survey was relatively comprehensive, not all determinants of implementation were assessed. For example, although we measured individual, innovation-related, and organizational constructs, we did not measure factors external to the school setting (i.e., district/state policies, partnerships with outside agencies) that may influence teachers’ intentions [66]. Further, we were faithful to the existing items and their scoring scales, which meant that some were 5- and some were 6-point scales and this may play a factor in interpretation. In addition, some items were slightly modified from adjacent topics (i.e., exercise benefits instead of CPA benefits) with wording changes that, although minimal, may have impacted validity. The analyses are cross-sectional and cannot assess causation. The intention item was not validated, although it is similar to validated measures [63,65] that ask a respondent to indicate agreement with statements such as “I intend to… (specific behaviors)”. Our item did not use the word “intend” so it may be argued that it does not represent behavioral intentions but rather a related construct such as behavioral plans. Finally, although we believe that more work is needed to explore intentions, we agree with others who note that intentions are not a perfect predictor of behavior and more theoretical and empirical work is needed to explore this disconnect [67].

## 5. Conclusions

This work uses a blended theoretical approach to explore individual and contextual factors associated with teachers’ CPA implementation intentions. We explored predictors of behavioral intentions, which are typically a strong predictor of future behavior. The biggest predictors of intentions to adopt CPA were compatibility, principal support, perceived autonomy, and openness to innovations. While much work has explored individual intentions to engage in health behavior change, less work has explored intentions regarding the implementation of innovations by intermediaries such as classroom teachers. Our work blends health behavior theory and implementation science [68], and highlights the role that intrinsic motivation—specifically, perceived autonomy—plays in teachers’ CPA implementation intentions.

## Figures and Tables

**Table 1 ijerph-20-03646-t001:** School and teacher demographic characteristics.

	Year 1			Year 2		Year 3				
	A	B	C	D	E	F	G	H	I	J
**School Characteristics** **(n = 10)**										
% White non-Hispanic/Latino students	70–80%	80–90%	40–50%	50–60%	40–50%	70–80%	60–70%	60–70%	60–70%	60–70%
% Hispanic/Latino students	10–20%	10–20%	50–60%	40–50%	40–50%	10–20%	20–30%	10–20%	30–40%	20–30%
% Students eligible for free/reduced-priced meals	40–50%	50–60%	80–90%	70–80%	80–90%	60–70%	50–60%	20–30%	60–70%	50–60%
Title 1 status	Yes	Yes	Yes	Yes	Yes	Yes	Yes	Yes	Yes	Yes
Enrollment	400–499	500–599	500–599	300–399	400–499	400–499	300–399	300–399	400–499	500–599
**Teacher Characteristics** **(n = 181)**										
Number of self-contained classroom teachers	18	22	22	16	20	18	15	12	17	21
Percentage female	94%	86%	100%	88%	95%	89%	100%	92%	88%	76%
Median number of years teaching	5 years	7 years	5 years	15 years	5 years	5 years	14 years	5 years	11 years	10 years
Range of years teaching	<1 to 39	<1 to 34	<1 to 16	1 to 35	1 to 20	<1 to 38	1 to 39	1 to 21	<1 to 21	<1 to 28
Number of early-career teachers (≤5 years)	9	9	11	4	10	9	5	6	6	3

Note. Each letter corresponds to a separate school; demographic information provided in ranges to preserve school anonymity.

**Table 2 ijerph-20-03646-t002:** Survey Constructs/Scales/Items, with Response Descriptive Data and Item-to-total correlations within Scales.

Construct/Items	Mean (SD)	Corrected Item-to-Total Correlation
**Relative Advantage (α = 0.888)**	5.06 (0.60)	
Providing opportunities for children to be physically active in my classroom would increase the quality of education my students receive.	5.10 (0.81)	0.662
Providing opportunities for children to be physically active in my classroom would enhance my effectiveness as a classroom teacher.	5.12 (0.75)	0.751
Providing opportunities for children to be physically active will increase my students’ academic performance more than teaching sedentary lessons with students at their desk.	5.20 (0.68)	0.682
Providing opportunities for children to be physically active in my classroom is consistent with my priorities as a teacher.	4.99 (0.69)	0.694
Providing opportunities for children to be physically active in my classroom fits well with the way I like to teach.	4.92 (0.83)	0.707
Providing opportunities for children to be physically active in my classroom is compatible with my educational philosophy.	5.05 (0.76)	0.807
**Perceived Barriers (α = 0.766)**	2.37 (0.68)	
My classroom environment can be easily modified to provide opportunities for my students to be physically active. *	1.90 (0.73)	0.461
My students make it easy for me to provide them with opportunities to by physically active in my classroom. *	2.25 (0.89)	0.474
The school schedule allows me to provide children with opportunities to be physically active in my classroom. *	2.60 (1.08)	0.551
The academic curriculum makes it hard for me to provide children with opportunities to be physically active in my classroom.	3.15 (1.29)	0.545
Providing opportunities for children to be physically active in my classroom requires more work than can be accomplished with current resources available to me.	2.16 (0.95)	0.530
Providing opportunities for children to be physically active in my classroom seems like just one more complication in an already busy schedule.	2.13 (0.97)	0.585
**Perceived Norms**	3.94 (1.06)	
Most other teachers at my school use activity breaks in their classrooms.	3.94 (1.06)	
**Principal Support for CPA**	5.40 (0.75)	
My principal would be supportive of my use of activity breaks in the classroom.	5.40 (0.75)	
**Perceived Competence (α = 0.894)**	5.34 (0.67)	
I feel confident in my ability to occasionally lead activity breaks in my classroom.	5.30 (0.83)	0.971
I am capable of using activity breaks in my classroom.	5.41 (0.68)	0.978
I believe I am able to achieve the goal of having a physically-active classroom.	5.29 (0.70)	0.970
**Perceived Autonomy for CPA (α = 0.743)**	5.31 (0.64)	
I feel free to choose which types of activity breaks to use, and when.	5.39 (0.70)	0.722
I have a say in choosing whether to use activity breaks.	5.20 (0.88)	0.659
I am in charge of how to incorporate physical activity into my classroom.	5.35 (0.78)	0.702
**Principal Autonomy Support (General)** **^a^ (α = 0.952)**	3.96 (0.92)	
My principal provides me choices and options.	3.96 (0.95)	0.760
I feel understood by my principal.	3.93 (1.04)	0.864
My principal conveys confidence in my ability to do well at my job.	4.13 (0.99)	0.863
My principal encourages me to ask questions.	4.03 (1.05)	0.855
My principal listens to how I would like to do things.	3.96 (1.04)	0.901
My principal tries to understand how I see things before suggesting a new way to do things.	3.79 (1.06)	0.871
**Openness to Educational Innovations (α = 0.743)**	4.38 (0.68)	
I adopt more new education ideas/classroom practices than other classroom teachers at my school.	3.71 (1.06)	0.578
If I learned that a new educational idea/classroom practice was available, I would be interested enough to consider adopting it.	4.84 (0.74)	0.433
In general, I am among the first of the classroom teachers at my school to know the latest trends in education/classroom teaching.	3.54 (1.06)	0.670
It is important to me to have the flexibility to try new things in my classroom.	**5.34 (0.71)**	0.496
**Organizational Climate: Collegiality** ** ^b^ ** **(α = 0.824)**	**4.05 (0.56)**	
Teachers respect the professional competence of their colleagues.	**4.13 (0.73)**	0.698
New teachers are readily accepted by colleagues.	**4.07 (0.79)**	0.501
Teachers are proud of their school.	**4.20 (0.71)**	0.664
Teachers help and support each other.	**4.17 (0.76)**	0.696
Teachers accomplish their work with vim, vigor, and pleasure.	**3.68 (0.70)**	0.568
**Organizational Climate: Restrictive Behavior** ** ^b^ ** **(α = 0.781)**	**2.91 (0.69)**	
Teachers are burdened with busy work.	**3.25 (0.87)**	0.564
Administrative paperwork is burdensome at this school.	**3.01 (0.86)**	0.655
Routine duties interfere with the job of teaching.	**2.78 (0.94)**	0.633
Teachers have too many committee requirements.	**2.65 (0.85)**	0.500
**Intention to Provide CPA**	**5.63 (0.54)**	
I definitely plan to try using activity breaks in my classroom.	**5.63 (0.54)**	

Note: All items scored from 1–6 (1 = strongly disagree to 6 = strongly agree) unless otherwise noted. ^a^ Responses scored from 1–5 (1 = not at all true, 5 = very true). ^b^ Responses scored from 1–5 (1 = never, 5 = very frequently). * Items were reverse-coded.

**Table 3 ijerph-20-03646-t003:** Mean Scores and Standard Deviations for Each Construct, by School.

School	Year 1			Year 2		Year 3					F
	A	B	C	D	E	F	G	H	I	J	
Number of teachers	18	22	22	16	20	18	15	12	17	21	
Predictor Variables											
Advantage/Compatibility	5.19 (0.55)	5.02 (0.41)	5.07 (0.68)	5.06 (0.51)	4.86 (0.79)	5.29 (0.58)	5.55 (0.43)	5.17 (0.66)	4.82 (0.47)	4.77 (0.59)	2.61 **
Perceived Barriers	2.67 (0.84)	2.38 (0.51)	2.17 (0.58)	2.64 (0.77)	2.60 (0.54)	2.01 (0.62)	2.04 (0.58)	2.29 (0.82)	2.38 (0.65)	2.46 (0.71)	2.26 *
Perceived Norms for CPA	4.00 (0.85)	4.86 (0.77)	4.19 (0.75)	3.33 (1.30)	3.89 (1.02)	4.28 (0.75)	4.47 (0.83)	3.91 (1.14)	3.38 (0.89)	2.74 (0.73)	8.93 ***
Principal Support for CPA	5.12 (1.05)	5.64 (0.58)	5.23 (0.62	4.69 (0.87)	5.30 (0.73)	5.78 (0.43)	5.67 (0.62)	5.83 (0.58)	5.47 (0.62)	5.40 (0.68)	4.04 ***
Perceived Competence	5.17 (0.85)	5.24 (0.65)	5.34 (0.96)	5.32 (0.89)	5.13 (0.73)	5.44 (0.51)	5.79 (0.36)	5.58 (0.57)	5.22 (0.39)	5.33 (0.65)	1.39
Perceived Autonomy for CPA	5.32 (0.83)	5.38 (0.64)	5.70 (0.58)	5.09 (0.63)	4.88 (0.65)	5.46 (0.50)	5.48 (0.60)	5.56 (0.57)	5.25 (0.44)	5.00 (0.57)	3.41 ***
Principal Autonomy Support (General) ^a^	3.43 (0.77)	4.30 (0.73)	3.59 (0.92)	3.12 (0.86)	3.82 (0.79)	4.71 (0.39)	4.14 (0.91)	4.42 (0.80)	4.49 (0.81)	3.72 (0.94)	6.98 ***
Perceived Norms for CPA	4.00 (0.85)	4.86 (0.77)	4.19 (0.75)	3.33 (1.30)	3.89 (1.02)	4.28 (0.75)	4.47 (0.83)	3.91 (1.14)	3.38 (0.89)	2.74 (0.73)	8.93 ***
Openness to Educational Innovations	5.13 (0.53)	5.06 (0.35)	4.97 (0.78)	4.96 (0.65)	4.80 (0.87)	5.28 (0.56)	5.48 (0.48)	5.00 (0.84)	4.69 (0.61)	4.61 (0.63)	2.68 **
Organizational Climate: Collegiality ^b^	4.02 (0.59)	4.44 (0.43)	4.06 (0.85)	3.72 (0.59)	3.79 (0.45)	4.33 (0.46)	4.36 (0.44)	4.12 (0.28)	4.2 (0.48)	3.48 (0.44)	7.48 ***
Organizational Climate: Restrictive Behavior ^b^	3.51 (0.72)	3.08 (0.60)	2.73 (0.55)	3.42 (0.70)	2.91 (0.62)	2.61 (0.63)	2.96 (0.56)	2.60 (0.60)	2.50 (0.70)	2.73 (0.59)	5.11 ***

Note: Responses scored from 1–6 (1 = strongly disagree, 6 = strongly agree) unless otherwise noted; ^a^ Responses scored from 1–5 (1 = not at all true, 5 = very true); ^b^ Responses scored from 1–5 (1 = never, 5 = very frequently); * *p* < 0.05, ** *p* < 0.01, *** *p* < 0.001.

**Table 4 ijerph-20-03646-t004:** Correlations between Constructs.

Construct	1.	2.	3.	4.	5.	6.	7.	8.	9.
1. Advantage/Compatibility									
2. Perceived Barriers	−0.45 ***								
3. Perceived Norms	0.29 ***	−0.19 *							
4. Principal Support for CPA	0.14	−0.31 ***	0.29 ***						
5. Perceived Competence	0.60 ***	−0.42 ***	0.16 *	0.32 ***					
6. Perceived Autonomy for CPA	0.46 ***	−0.43 ***	0.24 **	0.35 ***	0.55 ***				
7. Principal Autonomy Support (General)	0.14	−0.29 ***	0.29 ***	0.55 ***	0.25 **	0.24 **			
8. Openness to Educational Innovations	0.33 ***	−0.03	0.19 *	0.01	0.34 ***	0.22 **	0.03		
9. Organizational Climate: Collegiality	0.13 *	−0.14	0.47 ***	0.28 ***	0.13	0.26 **	0.44 ***	0.12	
10. Organizational Climate: Restrictive	0.10	0.21 **	−0.05	−0.28 ***	−0.12	−0.13	−0.43 ***	0.04	−0.19 *

* *p* < 0.05, ** *p* < 0.01, *** *p* < 0.001.

**Table 5 ijerph-20-03646-t005:** Results of Multilevel Logistic Regression Predicting Strong Teacher Intention to Implement CPA.

Construct	Odds Ratio	95% Confidence Interval	*p* Value
School-level poverty (% FRPM eligibility)			
<33% (referent)	1.00		
33 to <66%	1.01	0.42, 2.43	0.979
>66%	1.07	0.32, 3.57	0.921
Gender (1 = female)	2.84	0.42, 18.98	0.310
Grade taught			
Kindergarten to grade 2 (referent)	1.00		
Grades 3 to 6	1.29	0.32, 5.29	0.731
Number of years teaching			
<5 (referent)	1.00		
5–15	0.64	0.12, 3.40	0.614
>15	0.75	0.16, 3.41	0.714
Prior use of CPA			
Movement Breaks	1.14	0.44, 2.97	0.789
Active Lessons	0.76	0.10, 5.65	0.796
**Advantage/Compatibility**	**8.63**	**2.70, 27.61**	**0.005**
Perceived Barriers	0.35	0.12, 1.05	0.093
Perceived Norms	0.68	0.43, 1.08	0.134
**Principal Support for CPA**	**3.07**	**1.26, 7.51**	**0.036**
Perceived Competence	3.79	0.95, 15.14	0.092
**Perceived Autonomy for CPA**	**2.98**	**1.67, 5.32**	**0.005**
Principal Autonomy Support (General)	0.83	0.46, 1.47	0.536
**Openness to Educational Innovations**	**2.53**	**1.23, 5.21**	**0.032**
**Organizational Climate—Collegiality**	**0.53**	**0.34, 0.84**	**0.023**
Organizational Climate—Restrictive Behavior	2.14	0.44, 10.35	0.369

Note: Items in bold reflect significant predictor variables.

## Data Availability

De-identified datasets may be available by request sent to the principal investigator of the project, Dr. Lindsey Turner (lindseyturner1@boisestate.edu).

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
