# Peer review of "Individual and Contextual Factors Associated with Classroom Teachers’ Intentions to Implement Classroom Physical Activity"

_ijerph, 2023, doi:10.3390/ijerph20043646_

Round 1
Reviewer 1 Report
In my opinion, further measures should be specified in the theoretical framework (L110)
Except for misinterpretation, the entire theoretical framework focuses on other models that are not the ones finally used (Self Determination theory and additional measures). In addition, other potential determinants, not mentioned in the theoretical framework, are also cited.
There are some issues to be taken into account in the limitations such as the remuneration for participation or the appropriateness of the Likert scales in 5 and 6 items.
L135 in the first questionnaire 2 questions are mentioned, however 3 items are detailed
L147 The relationship of the variables chosen in the survey with respect to the models set out at the beginning of the article is not clear.
The extraction of some questions from other questionnaires does not guarantee that these questions are valid, as they may need the rest of the questions from the initial questionnaire to be valid and reliable. In addition, some changes in question wording are important and could lead to changes in the validity per se.
It would be necessary to express the goodness of fit of the multivariate regression.
The conclusion does not specify which variable is the most important predictor of intentions to implement CPA.
Reviewer 2 Report
Thank you for the Author's contribution to this study. The manuscript is well structured, but I would like to highlight a few points that need some improvement. These mostly include changes in the methods and the results. However, It must be highlighted that the topic is very important and the authors provide a full view of this topic and conclude relevant things.
· In the introduction the Authors introduce deeply what CPA is and how it can be associated with theoretical frameworks, however, the part with self-determination theory needs some improvements. It's not clear how SDT deals with CPA and other introduced frameworks. Please give me more detail about these associations.
· Please add more information about the measures. There is no answer to categorize in several places and it’s not entirely clear if the Authors used their own scales or they adapted one. Please be specific about this information in the methods.
· As I see most of the scales are previously used items that have references in the text. However, the Authors use factor analysis, but it's not clear why. I believe it would be enough to do a Multilevel Regression Analysis since all the used scales are previously validated. If the Author chooses to keep factor analysis please be specific about why was it important to use.
